# Sex Differences in Clinical Characteristics and Outcomes of Patients with SARS-CoV-2-Infection Admitted to Intensive Care Units in Austria

**DOI:** 10.3390/jpm12040517

**Published:** 2022-03-23

**Authors:** Alexandra Kautzky-Willer, Michaela Kaleta, Simon D. Lindner, Michael Leutner, Stefan Thurner, Peter Klimek

**Affiliations:** 1Department of Internal Medicine III, Clinical Division of Endocrinology and Metabolism, Medical University of Vienna, Währinger Gürtel 18-20, A-1090 Vienna, Austria; alexandra.kautzky-willer@meduniwien.ac.at (A.K.-W.); michael.leutner@meduniwien.ac.at (M.L.); 2Gender Institute, A-3571 Gars am Kamp, Austria; 3Section for Science of Complex Systems, CeMSIIS, Medical University of Vienna, Spitalgasse 23, A-1090 Vienna, Austria; kaleta@csh.ac.at (M.K.); simon.lindner@meduniwien.ac.at (S.D.L.); stefan.thurner@meduniwien.ac.at (S.T.); 4Complexity Science Hub Vienna, Josefstädter Strasse 39, A-1080 Vienna, Austria; 5Santa Fe Institute, 1399 Hyde Park Road, Santa Fe, NM 85701, USA

**Keywords:** sex, gender, diabetes, COVID-19, ICU, mortality

## Abstract

Importance: A male predominance is reported in hospitalised patients with COVID-19 alongside a higher mortality rate in men compared to women. Objective: To assess if the reported sex bias in the COVID-19 pandemic is validated by analysis of a subset of patients with severe disease. Design: A nationwide retrospective cohort study was performed using the Austrian National COVID Database. We performed a sex-specific Lasso regression to select the covariates best explaining the outcomes of mechanical ventilation and death using variables known before ICU admission. We use logistic regression to construct a sex-specific “risk score” for the outcomes using these variables. Setting: We studied the characteristics and outcomes of patients admitted to intensive care units (ICUs) in Austria. Participants: 5118 patients admitted to the ICU in Austria with a COVID-19 diagnosis in 03/2020–03/2021. Exposures: Demographic and clinical characteristics, vital signs and laboratory tests, comorbidities, and management of patients admitted to ICUs were analysed for possible sex differences. Main outcomes and measures: The aim was to define risk scores for mechanical ventilation and mortality for each sex to provide better sex-sensitive management and outcomes in the future. Results: We found balanced accuracies between 55% and 65% to predict the outcomes. Regarding outcome death, we found that the risk score for pre-ICU variables increases with age, renal insufficiency (f: OR 1.7(2), m: 1.9(2)) and decreases with observance as admission cause (f: OR 0.33(5), m: 0.36(5)). Additionally, the risk score for females also includes respiratory insufficiency (OR 2.4(4)) while heart failure for males only (OR 1.5(1)). Conclusions and relevance: Better knowledge of how sex influences COVID-19 outcomes at ICUs will have important implications for the ongoing pandemic’s clinical care and management strategies. Identifying sex-specific features in individuals with COVID-19 and fatal consequences might inform preventive strategies and public health services.

## 1. Introduction

The coronavirus disease 2019 (COVID-19) outbreak resulted in pandemic and far-reaching economic, political, and healthcare-related changes, influencing everyone’s lives. In addition, the novel disease caused by the severe acute respiratory syndrome coronavirus 2 (SARS-CoV-2) evidenced important differences between men and women in regard to infection rates, admissions to hospitals, need for intensive care, including respiratory support, mortality, and long-term outcomes [1,2,3]. Nevertheless, sex-specific clinical data and national analysis stratified by sex are scarce in the scientific COVID-19 related literature. Especially clinical management of hospitalised patients and information on specific therapies are largely missing.

Therefore, the aim of this study was to analyse differences between men and women admitted to intensive care units (ICUs) in Austria, a high-income Central European country with a rather homogeneous population with general healthcare insurance coverage. However, a recent analysis reported rather large gender differences in a multidimensional ageing Index, accounting for differences in societal ageing in Austria, favouring men in several domains, including productivity, engagement, security and cohesion [4]. Although women have higher life expectancy in Austria (83.7 years vs. 78.9 years in 2020) like in most other European countries, healthy life expectancy appears to be similar between men and women.

In general, men appear to be at higher risk of severe or fatal COVID-19 disease irrespective of age, region and despite comparable infection rates in both sexes. This may be ascribed to the combined protective effects of X-linked immune response genes and the female sex steroid hormones enhancing antibody production and mitigating innate immune inflammatory responses [5,6]. However, better compliance with preventive measures, healthier lifestyles and lower rates of unfavourable comorbid conditions in women compared to men may influence gender differences in risk and outcome [7]. On the other hand, the situation may differ in severe forms, requiring hospitalization or even intensive care. As severe COVID-19 especially affects older people with comorbidities, we were interested to see if there are differences in age, comorbidities, duration of ICU stays, the necessity of mechanical ventilation and mortality between sexes. Therefore, our objective was to evaluate factors related to severe disease and unfavourable outcomes of patients admitted to ICUs with a particular focus on possible sex differences. Identifying clinical characteristics and comorbid conditions associated with a high risk of ICU admission, mechanical ventilation, and death, stratified by sex, may help develop sex-sensitive and more personalized prevention and treatment strategies.

## 2. Methods

The analysed dataset contains detailed information on 36,188 Austrian COVID-19 patients in hospital care between 03/2020–03/2021, out of which 5118 patients were admitted to ICU. The dataset combines three different administrative datasets, namely hospital claims (XDok) with two standardized ICU datasets (“Therapeutical Intervention Scoring System” TISS-A and “Simplified Acute Physiology Score” SAPS-3) into an anonymized research dataset containing all patients hospitalized in Austria in the mentioned time span that have a main or side diagnosis of COVID-19. The data can be accessed by research institutions via the data platform of the Austrian National Public Health Institute via an accreditation procedure. Information in these datasets (including transfer reason to ICU) is coded by clinicians as part of the federal hospital performance documentation. We consider general characteristics (age, sex, hospital duration), causes of admission, transfer type, comorbid conditions, treatment, vital signs and Glasgow Coma Scale, and laboratory values. We group these variables into those known before ICU admission; see Table 1; and those recorded during the ICU stay; see Table 2.

The outcome variables are death and mechanical ventilation. We use Lasso regression to select a specified number of covariates (n) for each sex and the pre-ICU variable set (Table 1) by variation of the regularisation strength. For robustness, the feature selection is repeated 100 times with a 70/30 train-test data split. The final covariates (n) are the features that are selected most frequent by this process.

To obtain estimates for the combined effect of the selected covariates (“risk scores”), we divide our data into holdout, test and training sets (100 holdout sets with each 100 test/training splits) and use logistic regression to assess effect sizes of the selected covariates. We perform sub-analyses for each outcome of interest (two binomial outcomes, mechanical ventilation and death). The regressions are sex-specific, meaning separate input data for females and males, for both outcomes using Python (pandas, sklearn). We compute the models’ quality measures (specificity, sensitivity and balanced accuracy) using the averaged regression predictions both on the holdout and the test dataset.

We perform several lasso regressions for different numbers of selected covariates (from *n* = 2 to *n* = 9) to compare the model’s quality measures. Additionally, we use the lasso regression to determine the best fitting number of covariates *n* by analysing the distribution of selected features over the cross-validation. We found the smallest variation in selected variables for *n* = 5. In addition, the final resulting logistic regression quality measures (specificity, sensitivity and balanced accuracy) appear to be robust under variation of number of selected covariates. A higher number of selected covariates does not lead to an increase in model quality.

This research project was approved by the Austrian COVID-19 Data Platform under #02/2020.

## 3. Results

In our national Austrian cohort, 36,188 patients (48% females) were hospitalised and 5118 patients (35% females) required ICU management. 3499 (85%) of ICU patients had at least one comorbidity (females: 85%, males: 86%). The Charlson Comorbidity Index was 1.4 (SD 2.3) for all patients and somewhat higher in females than males (1.6 [SD 2.4] vs. 1.4 [SD 2.3]).

Baseline characteristics are shown in Table 1 for pre-ICU variables and Table 2 for variables recorded during the ICU stay. Regarding the pre-ICU variables, we found that males were more likely to be admitted because of respiratory causes (specific cause: 66% of females [f] vs. 74% of males [m], *p* < 0.0001; general cause: 35% [f] vs. 42% [m], *p* < 0.0001) whereas females were more likely to be admitted for observance (10% [f] vs. 7% [m], *p* < 0.001). Regarding comorbid conditions, females were more likely to have renal insufficiency (19% [f] vs. 14% [m], *p* < 0.001) and males more likely to have respiratory infections (48% [f] vs. 53% [m], *p* < 0.001) and type 2 diabetes. There were no significant sex differences in the transfer to ICU except for transfer from surgery which was found in a higher proportion of women.

During the stay, males had longer ICU stay durations (mean [SD] of 9 [11, f] vs. 11 [13, m] days, *p* < 0.0001) and longer total hospital stay durations (10 [13, f] vs. 12 [16, m] days, *p* < 0.0001). Males had higher temperature (37.1 [1.2, f] vs. 37.3 [1.2, m], *p* < 0.0001) and oxygen concentration. Regarding the Glasgow Coma Scale, males were more likely to open their eyes spontaneously as eye response (72% [f] vs. 76% [m], *p* < 0.0001), to be oriented and converse normally in their verbal response (68% [f] vs. 71% [m], *p* < 0.001) and to obey commands in their motor response (74% [f] vs. 78% [m], *p* < 0.0001). Furthermore, males had higher Bilirubin (0.72 [1.4, f] vs. 0.87 [2.1, m] mg/dL, *p* < 0.0001) and Creatinine (1.4 [1.8, f] vs. 1.5 [8, m] mg/dL, *p* < 0.0001) and arterial PH values. In addition, laboratory values revealed lower minimum leucocytes in women and lower minimum thrombocytes in men.

Men experienced artificial feeding more often, both enteral and parenteral nutrition, compared to women in ICUs. Resuscitation more often occurred in men and they were more often in need of tracheal tubes, whereas women were more often supplied with oxygen masks.

The resulting risk scores (regression coefficients) for the different outcomes and covariates are shown in Table 3, Table 4 and Table 5. In the following, we only report those risk factors that increased (decreased) outcome risk by more than 5%. Regarding outcome death, we found that the risk score for pre-ICU variables increases with age (f: OR 1.062(5), m: 1.066(4)), renal insufficiency (f: OR 1.7(2), m: 1.9(2)) and decreases with observance as admission cause for males and females (f: OR 0.33(5), m: 0.36(5)); see Table 3. Additionally, the risk score for females also includes respiratory insufficiency (OR 2.4(4)) while heart failure for males only (OR 1.5(1)).

Regarding mechanical ventilation, the risk score for pre-ICU variables decreased with transfer from inpatient ward (f: OR 0.67(6), m: OR 0.45(3)) whereas increased with respiratory infections (f: OR 1.9(2), m: OR 1.6(1)) and admittance for respiratory reasons (f: OR 1.7(2), m: OR 1.6(1)) for both sexes, as shown in Table 4. For males, the risk score decreased strongly being transferred from emergency rooms (OR 0.36(4)) whereas for females with admittance for observance (OR 0.68(6)).

Regarding the predictive value of these risk scores, see Table 5. We obtained balanced accuracies in the range of 63–65% for outcome death and 56–58% for outcome mechanical breathing. Sensitivities ranged from 40–50% for death and around 25% for mechanical breathing.

## 4. Discussion

In our national Austrian cohort, almost half of the hospitalised patients were females, but they only comprised 35% of the patients requiring ICU management. However, the mortality rate of 45% in females and 44% in males did not show sex disparities among the ICU patients. The global COVID-19 sex-disaggregated data tracker reports for Austria a rather sex-balanced number of confirmed cases and that 53% of the overall COVID-19 related mortality cases affected males [8]. Here, women were older, which is consistent with most other studies. Male ICU patients showed longer both hospital and ICU duration time compared to females. Of note, it was previously reported that male sex was associated with receiving more ICU care per admission, using more ICU resources [9].

In our study, the majority of the patients presented with at least one comorbidity without noteworthy sex differences. However, the Charlson Comorbidity Index was slightly but significantly higher in females than males. We also observed sex differences in clinical characteristics, comorbidities, treatments and outcomes, and differences in risk predictors of death and IMV in men and women.

Differences between men and women were evidenced in hospitalised patients, mostly from US metropolitans. A case series of patients hospitalised with COVID-19 in the New York City Area (40% women) showed that the most frequently described comorbidities were hypertension, obesity and diabetes [10]. 88% had more than one comorbidity but the median Charlson Comorbidity was much higher than in our cohort, reaching 4. Fourteen percent were in need of intensive care, 12% received IMV and 21% died. Mortality rates were higher for men than women at all decades older than 20 years.

A series of patients in Metropolitan Detroit showed that—in contrast to other reports—56% of treated patients were female, the majority being African Americans, which may be ascribed to the high poverty rate and thus susceptibility of this subgroup. In line with our study, most patients had at least one comorbidity, including hypertension, chronic kidney disease (CKD) and diabetes. 76% were hospitalised and 40% required ICU management during their stay [11]. Similar to our data, males comprised 57% of the ICU cohort. Male sex, obesity and CKD, each doubled the risk of ICU treatment. The mortality rate of ICU-managed patients was 40%, which corresponds to our data. Male sex and age older than 60 years were independently related to mortality and IMV. Obesity, CKD and cancer also related to more than doubled the risk of IMV. However, as a limitation, we acknowledge that our data contained no information from which a potential minority status could have been inferred, meaning that it remains to be seen how, e.g., ethnicity influenced our results.

Another sex-specific analysis of hospitalized COVID-19 patients in New Orleans (61% women) revealed that rates of ICU, IMV and in-hospital mortality were comparable between sexes, which is in line with our results but in stark contrast to most other reports [12].

The authors also highlighted sex disparities in clinical determinants of severe outcomes. Interestingly, obesity, diabetes and hypertension were more prevalent in women and diabetes, CKD, and an increased neutrophile-to-lymphocyte ratio and ferritin levels independently predicted death in women only. Interesting results could also be observed in an observational retrospective cohort study conducted in Germany (*n* = 23,235 patients) where it has been shown that once ventilated, the advantage of females in survival seems to disappear [13]. However, there are some important differences: in the German ICU cohort, 65% of the males but only 35% of the females needed ventilation therapy. In our cohort, there was no sex difference, with a similar number of women (36%) and men (38%) in need of mechanical ventilation. In our cohort, mechanical ventilation did not change the sex ratio of mortality risk. Moreover, we did not see the divergence of curves in men vs. women after the age of 60 years. As both studies are retrospective, observational studies both lack some important information limiting the possibility of direct comparisons. Therefore, differences in patients’ clinical characteristics or psychosocial factors appear to be responsible for the controversial findings.

The Italian SARS-RAS study reported data of 395 ICU patients, mostly men (74%), with a higher prevalence of comorbidities [14]. The main predictors of ICU admission were male sex, obesity, CKD, and hypertension in the total cohort. Sex-disaggregated analysis showed that these variables remained predictors among men, whereas in women, heart failure next to obesity is associated with higher ICU admission rates. Of note, despite the small Italian ICU sample size compared to our cohort, they also found a higher prevalence of heart failure in women than men. However, because heart failure predicted death only in male ICU patients in our cohort, it may be hypothesized that heart failure is a greater risk factor for ICU admission in infected females but for mortality in male ICU patients.

Sex-differentiated analysis of the clinical phenotype and transitions of care among individuals dying of COVID-19 in another Italian cohort showed that at hospital admission, men had a higher prevalence of ischemic heart disease, chronic obstructive pulmonary disease, and CKD, while women were older and more likely to have dementia and autoimmune disease [15]. However, both sexes had a high level of multimorbidity.

Hypertension appears to be the most common comorbidity in various studies, particularly in those admitted to hospitals. This was also evident in our analysis without sex differences. Obesity also seems to be a very common comorbid condition with an increased risk of severe disease. Unfortunately, BMI or weight class is not available in our data set. However, we suppose that also in our cohort, a substantial number of patients were overweight/obese as the presence of T2DM, hypertension, heart failure, or CKD, conditions commonly related to the metabolic syndrome, was rather high among our ICU patients and comparable to many other studies. 

In addition, in accordance with other studies, dyspnoea and respiratory problems were the main specific causes of ICU admission and the need for IMV in our study [11,16]. However, respiratory problems, including acute lung injuries and acute respiratory distress syndrome, more often caused admission in men compared to women. These causes of transfer to ICU are also related to increased death rates in both men and women with slightly greater impact in women. A greater percentage of women was transferred from surgery, whereas no sex differences were seen in other transfers to ICUs. Additionally, females more often had a general cause for ICU-admission due to “Observance, post-operative, and post-interventional monitoring”. These data indicate that there could be a higher risk for severe postoperative complications in females, necessitating ICU.

Moreover, post-operative or interventional monitoring more frequently induced ICU admission in infected women than men.

A meta-analysis revealed that combined mortality rates were 60% at the beginning of the pandemic and declined to 42% over time, ranging from 0 up to 85% [17]. The male bias towards severe COVID-19 disease appears to be a consistent feature throughout the pandemic [18]. Although there is evidence from the global COVID-19 meta-analysis that male sex is associated with an almost threefold higher risk of being admitted to an ICU and higher odds of death compared to females [18], the picture among SARS-CoV-2-infected ICU patients is inconsistent.

Analysis from retrospective cohort studies of critically ill patients admitted to ICUs in Northern Italy, including 25% women, confirmed that most of these patients were in need of IMV, and approximately half of them died in the ICUs [19]. Multivariate analysis revealed that age (HR 1.75%) and male sex (HR 1.57%) were significantly associated with mortality. Among comorbidities, only hyperlipidemia, diabetes, and COPD were significantly related to mortality. No medication is independently associated with mortality. Sex-stratified analyses were not reported.

A systematic review and meta-analysis searched for factors related to death among patients admitted to ICUs [20]. Interestingly, despite robust risk estimations, sex and increasing BMI were no significant risk factors, whereas older age, smoking, hypertension, diabetes, CVD, respiratory disease, CKD, and cancer were all associated with mortality. The highest odds were seen for mechanical ventilation at admission. In our study, women featured more often heart failure and renal insufficiency, whereas men more frequently showed respiratory infections or type 2 diabetes as comorbid conditions. Searching for pre-ICU variables related to higher death rates, we confirmed that older age and renal insufficiency/CKD are associated with death in both sexes, whereas heart failure is only in men, but respiratory insufficiency is only in women. Interestingly women were shown to have an increased propensity to develop ARDS in previous studies in critically ill patients [21]. On the other hand, ICU admission based on post-operative and interventional monitoring reduced mortality risk in both sexes.

Searching for the possible effect of pre-ICU variables on IMV, we found that only respiratory infections and complications impacted this outcome in both men and women with somewhat greater effects in women. Older age-related to slightly lower risk probably because of bias according to triage concepts. Transfer from the emergency department related to greatly reduced risk in men while a transfer for post-interventional monitoring slightly decreased risk of IMV in women only.

Diabetes was not independently related to IMV or death in our study, neither in men nor women. Other recent studies also showed that in patients with diabetes, the male sex bias vanished. In COVID-19, especially women with type 2 diabetes appear to lose their female biological advantage resulting in comparable death rates to those of men [22,23]. A multicentre study from Austria, including hospitalised COVID-19 patients with prediabetes, DM1, or DM2, found that neither sex nor BMI were risk factors for mortality [24].

Men presented at ICU with better performance in the Glasgow Coma Scale, showing more often spontaneous eye responses and oriented verbal as well as precise motor responses. Regarding these measurements, presentation of vital signs, available laboratory parameters, or treatments at ICU, we found some sex-dimorph data as shown in Table 2, but these variables were not selected amongst the most explanatory variables regarding outcome parameters.

Overall, we found these variables to be poor predictors of the outcomes, with sensitivities of around 25% in the risk prediction for mechanical breathing and between 40–50% for mortality. This means that only around one-fourth to half of these outcomes could have been predicted based on the available information at admission. We conducted several analyses that, for instance, used more or fewer variables or also included information recorded already during the stay but did not find a substantial increase in model quality during these analyses.

Therefore, in light of these findings as well as of existing controversial results [14,18,25,26,27], more research is necessary to further explore associations between sex and mortality and to perform sex-disaggregated analysis, especially among ICU patients. The gap in the literature highlights that sex is still an underappreciated parameter when requesting outcomes of patients with confirmed COVID-19 disease. However, better knowledge of sex-specific risks and possible disparities is important for precision medicine and the continuing public health response to this pandemic.

## Figures and Tables

**Table 1 jpm-12-00517-t001:** Comparison of patients with Coronavirus Disease 2019 admitted to the intensive care unit (ICU), by sex. Pre-ICU characteristics. The number of patients with observed characteristics for categorical variables or mean value for continuous variables are shown as well as the relative numbers (%) or standard deviation (SD) in parentheses. *p*-value significance tests between values of both sexes are performed using Chi-squared test for categorical variables or *t*-test for continuous variables.

	Mean or No. (%)			
	All (N = 5118)	Females (N = 1796)	Males (N = 3322)	F/M *p*-Value
Sex [%]		35%	65%	
Age (SD)	66 (15)	68 (15)	65 (14)	<1 × 10^−5^
Missing data [%]	1022 (20)	407(23)	615(19)	<0.001
** *Specific cause for ICU-Admission* **				
Gastrointestinal	194 (5)	71 (5)	123 (5)	0.65
Cardiovascular	581 (14)	216 (16)	365 (13)	0.26
Liver disease	97 (2)	29 (2)	68 (3)	0.28
Metabolic/endocrine	277 (7)	107 (8)	170 (6)	0.20
Neurological	353 (9)	133 (10)	220 (8)	0.29
Renal	256 (6)	97 (7)	159 (6)	0.34
Respiratory	2930 (72)	919 (66)	2011 (74)	<1 × 10^−5^
Severe trauma	41 (1)	19 (1)	22 (1)	0.13
Haematological	41 (1)	22 (1)	19 (1)	0.12
** *General cause for ICU-Admission* **				
Observance, post-operative, and post-interventional monitoring	324 (8)	142 (10)	182 (7)	<0.001
Respiratory, ALI ^1^ and ARDS ^2^	1627 (40)	482 (35)	1145 (42)	<1 × 10^−5^
Observance, monitoring	2351 (57)	782 (56)	1569 (58)	0.012
** *Transfer to ICU* **				
Transfer from IMCU ^3^	228 (6)	71 (5)	157 (6)	0.20
Transfer from ER ^4^	667 (16)	233 (17)	434 (16)	0.93
Transfer from surgery	258 (6)	109 (8)	149 (6)	0.012
Transfer from other ICU	493 (12)	162 (12)	331 (12)	0.27
Transfer from inpatient ward	2090 (41)	689 (38)	1401 (42)	0.090
** *Comorbid Conditions* **				
Heart failure	863 (21)	320 (23)	543 (20)	0.027
Respiratory insufficiency	362 (9)	129 (9)	233 (9)	0.47
Renal insufficiency	644 (16)	260 (19)	384 (14)	<0.001
COPD ^5^	532 (13)	176 (13)	356 (13)	0.67
Immunosuppression	115 (3)	41 (3)	74 (3)	0.69
Respiratory infection	2108 (51)	664 (48)	1444 (53)	<0.001
Hematological Disease	113 (3)	44 (3)	69 (3)	0.25
T2DM ^6^	678 (17)	205 (15)	473 (17)	0.027
T1DM ^7^	327 (8)	122 (9)	205 (8)	0.18
Metastatic cancer	80 (2)	31 (2)	49 (2)	0.36
Non-metastatic cancer	167 (4)	47 (3)	120 (4)	0.11
Liver cirrhosis	63 (2)	19 (1)	44 (2)	0.53
Hypertension	2363 (57)	820 (59)	1543 (57)	0.21
Any condition	3499 (85)	1183 (85)	2316 (86)	0.0044
Charleson Comorbidity Score	1.4 (2.3)	1.6 (2.4)	1.4 (2.3)	0.017

1 Acute Lung Injury; 2 Acute Respiratory Distress Syndrome; 3 Intermediate Care Unit; 4 Emergency room; 5 Chronic obstructive pulmonary disease; 6 Type 2 diabetes mellitus; 7 Type 1 diabetes mellitus.

**Table 2 jpm-12-00517-t002:** Comparison of patients with Coronavirus Disease 2019 admitted to the intensive care unit (ICU) by sex. ICU characteristics. The number of patients with observed characteristics for categorical variables or mean value for continuous variables are shown as well as the relative numbers (%) or standard deviation (SD) in parentheses. *p*-value significance tests between values of both sexes are performed using Chi-squared test for categorical variables or *t*-test for continuous variables.

	Mean or No. (%)			
	All (N = 5118)	Females (N = 1796)	Males (N = 3322)	F/M *p*-Value
Missing data [%]	1022 (20)	407(23)	615(19)	<0.001
** *ICU stay* **				
ICU duration time (SD)	11 (12)	9 (11)	11 (13)	<1 × 10^−5^
Hospital duration time (SD)	11 (14)	10 (13)	12 (16)	<1 × 10^−5^
Death	1819 (44)	623 (45)	1196 (44)	0.35
** *Treatment* **				
Parenteral Nutrition	1223 (30)	387 (28)	836 (31)	0.045
Enteral nutrition	1053 (26)	330 (24)	723 (27)	0.040
Oxygen mask	1273 (31)	468 (34)	805 (30)	<0.01
Tracheal tube	1086 (27)	354 (25)	732 (27)	0.28
Central venous catheter	2653 (65)	912 (66)	1741 (64)	0.40
Vasoactive drugs pre-ICU	677 (17)	237 (17)	440 (16)	0.51
Steroids	197 (5)	76 (5)	121 (4)	0.16
Mechanical Ventilation	1543 (38)	502 (36)	1041 (38)	0.15
CPAP ^8^	1142 (28)	380 (27)	762 (28)	0.59
Tracheal Cannula	191 (5)	53 (4)	138 (5)	0.065
Resuscitation	68 (2)	15 (1)	53 (2)	0.037
Delirium	3082 (75)	1046 (75)	2036 (75)	0.96
Severe metabolic dysfunction	161 (4)	54 (4)	107 (4)	0.92
** *Vital Signs* **				
Temperature (SD) [°]	37.3 (1.2)	37.1 (1.2)	37.3 (1.2)	<1 × 10^−4^
Min. systolic blood pressure (SD) [mmHg]	112 (36)	112 (37)	112 (36)	0.97
Maximum heart rate (SD) [bpm]	94 (25)	94 (24)	95 (25)	0.18
Oxygen concentration (SD) [PaO_2_]	62 (24)	60 (24)	62 (24)	<0.01
** *Glasgow Coma Scale* **				
Eye response—Does not open eyes	312 (8)	125 (9)	187 (7)	0.058
Eye response—Opens eyes in response to pain	148 (4)	59 (5)	89 (3)	0.22
Eye response—Opens eyes in response to voice	341 (8)	132 (10)	209 (8)	0.15
Eye response—Opens eyes spontaneously	3065 (75)	1002 (72)	2063 (76)	<1 × 10^−4^
Eye response—Unknown	230 (6)	71 (5)	159 (6)	0.17
Verbal response—Makes no sounds	380 (9)	157 (11)	223 (8)	<0.01
Verbal response—Makes sounds	95 (2)	43 (3)	52 (2)	0.036
Verbal response—Words	71 (2)	30 (2)	41 (2)	0.20
Verbal response—Confused, disoriented	389 (10)	123 (9)	266 (10)	0.14
Verbal response—Oriented, converses normally	2881 (70)	950 (68)	1931 (71)	<0.001
Verbal response—Unknown	280 (7)	86 (6)	194 (7)	0.11
Motor response—No motor response	302 (7)	120 (9)	182 (7)	0.081
Motor response—Flexion/Withdrawal to painful stimuli	113 (3)	51 (4)	62 (2)	0.024
Motor response—Localizes to painful stimuli	274 (7)	99 (7)	175 (6)	0.71
Motor response—Obeys commands	3137 (77)	1024 (74)	2113 (78)	<1 × 10^−5^
Motor response—Unknown	248 (6)	80 (6)	168 (6)	0.34
** *Laboratory values* **				
Minimum leukocytes (SD) [g/L]	11 (10)	10 (11)	11 (9)	<0.01
Minimum thrombocytes (SD) [g/L]	227 (107)	236 (115)	222 (102)	<0.01
Bilirubin (SD) [mg/dL]	0.82 (1.9)	0.72 (1.4)	0.87 (2.1)	<1 × 10^−5^
Creatinine (SD) [mg/dL]	1.4 (1.8)	1.4 (1.8)	1.5 (8)	<1 × 10^−5^
PH values (SD) [g/L]	7.392 (0.112)	7.387 (0.112)	7.394 (0.112)	<0.01

8 Continuous positive airway pressure.

**Table 3 jpm-12-00517-t003:** Pre-ICU covariates for outcome “death”. Table shows case numbers in the sub-cohorts as well as the averaged regression coefficients for the selected features.

	Mean or No. (%)			Risk Score (SD)	
	Females (N = 318)	Males (N = 666)	F/M *p*-Value	Females	Males
Age (SD)	73 (11)	70 (11)	<0.001	1.0620 (0.0051)	1.0659 (0.0038)
Respiratory insufficiency	42 (13)	-	-	2.40 (0.37)	-
Renal insufficiency	85 (27)	137 (21)	0.011	1.69 (0.21)	1.88 (0.18)
Heart failure	-	185 (28)	-	-	1.49 (0.12)
AC: Observance, post-operative, and post-interventional monitoring	16 (5)	23 (3)	0.40	0.327 (0.050)	0.363 (0.054)

**Table 4 jpm-12-00517-t004:** Pre-ICU covariates for outcome “mechanical ventilation”. Table shows case numbers in the sub-cohorts as well as the averaged regression coefficients for the selected features.

	Mean or No. (%)			Risk Score (SD)	
	Females (N = 398)	Males (N = 812)	F/M *p*-Value	Females	Males
Age (SD)	66 (14)	64 (14)	<e^−4^	0.99350 (0.00031)	0.9904 (0.0023)
Respiratory infection	240 (60)	499 (61)	0.052	1.92 (0.18)	1.64 (0.11)
Transfer from ER	-	84 (10)	-	-	0.356 (0.035)
Transfer from inpatient ward	167 (42)	361 (44)	0.54	0.667 (0.059)	0.446 (0.031)
AC: Observance, monitoring	198 (50)	-	-	0.684 (0.063)	-
AC: Respiratory, ALI and ARDS	176 (44)	396 (49)	0.052	1.74 (0.16)	1.579 (0.098)

**Table 5 jpm-12-00517-t005:** Averaged model quality based on 100 holdout data splits. Shows specificity (true positive rate), sensitivity (true negative rate) and balanced accuracy for all evaluated regression models.

	Specificity (SD)	Sensitivity (SD)	Balanced Accuracy (SD)
Death			
Males	0.812 (0.023)	0.496 (0.043)	0.654 (0.020)
Females	0.849 (0.039)	0.407 (0.049)	0.628 (0.021)
Mechanical breathing			
Males	0.889 (0.022)	0.262 (0.032)	0.575 (0.016)
Females	0.879 (0.037)	0.252 (0.049)	0.566 (0.023)

## Data Availability

The data can be made available to accredited institutions via the Austrian COVID-19 Data Platform accessible under https://datenplattform-covid.goeg.at/.

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
