# Peer review of "Sex Differences in Clinical Characteristics and Outcomes of Patients with SARS-CoV-2-Infection Admitted to Intensive Care Units in Austria"

_jpm, 2022, doi:10.3390/jpm12040517_

Round 1

Reviewer 1 Report

The authors present data from a national COVID-29 registry/cohort. In the abstract it is called a National Database. This does possibly not include all patients in Austria? Please describe this registry, selection process (selection bias as a limitation?), informed consent, data structure, GDPR adherence and vote of an ethics committe.

  As the registry is not quite clear one may wonder, how was e.g. reason for transfer to ICU evaluated? If this is prospective registry, was this decided/answered by a nurse, a clinician? Could there be several reasons? The gender difference in this item is astounding and deserves an explanation.  

The authors describe that there is no longer a femal sex advantage in terms of mortality once patients are on ICU. This issue is elaborated in more detail by Nachtigall et al, as described by Nachtigall et al, 2021. It seems, that the female sex advantage disappears after invasive ventilation and thus looking simply at the factor "ICU stay" obscures the differences among different disease severities on ICU. In addition, data published by Nachtigall et al: Sex Differences in Clinical Course and Intensive Care Unit Admission in a National Cohort of Hospitalized Patients with COVID‐19. J. Clin. Med. 2021, 10, 4954. are of similar cohort size and originated from Germany and offer interesting comparison.

Reviewer 2 Report

I have read with interest the manuscript Sex differences in clinical characteristics and outcomes of patients with SARS-Cov2-infection admitted to intensive care units in Austria. Although controversial results have been reported in literature regarding sex and mortality in COVID-19 I find the research design appropriate. 

I would kindly ask the authors to include in introduction the exact life expectancy for Austrian population. 

Were there patients of minority included in your study? If yes please discuss in regard with the series of patients in Metropolitan Detroit you mentioned.

Please insert in your manuscript the approval number of the local Ethics Committee. 

Round 2

Reviewer 1 Report

Thank you for answering queries and accepting recommendations.